# Wide range of possible trajectories of North Atlantic climate in a warming world

Qinxue Gu [1] ✉, Melissa Gervais [1,2], Gokhan Danabasoglu [3], Who M. Kim [3], Frederic Castruccio [3], Elizabeth Maroon [4] & Shang-Ping Xie [5]

Decadal variability in the North Atlantic Ocean impacts regional and global climate, yet changes in internal decadal variability under anthropogenic radiative forcing remain largely unexplored. Here we use the Community Earth System Model 2 Large Ensemble under historical and the Shared Socio-economic Pathway 3-7.0 future radiative forcing scenarios and show that the ensemble spread in northern North Atlantic sea surface temperature (SST) more than doubles during the mid-twenty-first century, highlighting an exceptionally wide range of possible climate states. Furthermore, there are strikingly distinct trajectories in these SSTs, arising from differences in the North Atlantic deep convection among ensemble members starting by 2030. We propose that these are stochastically triggered and subsequently amplified by positive feedbacks involving coupled ocean-atmosphere-sea ice interactions. Freshwater forcing associated with global warming seems necessary for activating these feedbacks, accentuating the impact of external forcing on internal variability. Further investigation on seven additional large ensembles affirms the robustness of our findings. By monitoring these mechanisms in real time and extending dynamical model predictions after positive feedbacks activate, we may achieve skillful long-lead North Atlantic decadal predictions that are effective for multiple decades.

Accurate climate prediction on timescales ranging from years to decades provides invaluable information for climate adaptation and resilience, thus facilitating informed decision-making among governmental agencies and industry sectors in response to the rapidly changing climate[1]. Assessing the reliability of such predictions requires a thorough understanding of the mechanisms driving climate variability on decadal timescales[2,3]. Sources of decadal variability encompass anthropogenic external forcing (e.g., greenhouse gas emissions), natural external forcing (e.g., volcanic eruptions), and internal variability. Among these sources, internal variability can lead to large uncertainties in decadal predictions[4–8], with such uncertainties possibly evolving over time due to changes in external forcing[9–13].

The North Atlantic Ocean exhibits prominent interannual-to-multidecadal variability, as documented in both observational[14–17] and modeling studies[18–20]. This variability influences regional and global climate, such as North American heat waves[21], North Atlantic tropical cyclone activity[22], Arctic sea ice extent[23], Asian monsoon patterns[24], and precipitation worldwide[25–28]. Numerous studies have delved into understanding the mechanisms that govern North Atlantic sea surface temperature (SST) variability[29–35]. However, what remains an open question is how anthropogenic external forcing may modulate internal variability in the North Atlantic on decadal timescales.

With anthropogenic emissions such as greenhouse gases and aerosols, the mean state of the North Atlantic can experience rapid

[1]Department of Meteorology and Atmospheric Science, The Pennsylvania State University, University Park, PA, USA. [2]The Institute for Computational and Data Sciences, The Pennsylvania State University, University Park, PA, USA. [3]National Science Foundation National Center for Atmospheric Research, Boulder, CO, USA. [4]Department of Atmospheric and Oceanic Sciences, University of Wisconsin-Madison, Madison, WI, USA. [5]Scripps Institution of Oceanography, University of California San Diego, La Jolla, CA, USA. ✉e-mail: qzg18@psu.edu

changes. One notable example is the projected weakening of the Atlantic meridional overturning circulation (AMOC) in a warming climate, which can be attributed to increased heat and freshwater flux into the ocean, and thus a reduction or cessation of North Atlantic deep convection[36–39]. The AMOC is a key driver of poleward heat transport in the North Atlantic Ocean[40–42]. Therefore, the weakened AMOC has various climate impacts, including the so-called North Atlantic warming hole[38,43–46] and its related atmospheric impacts[47–49], which vary with AMOC decline rates in different model simulations[50]. Given these mean changes under external forcing, it is crucial to examine how North Atlantic internal variability may evolve in response.

In this study, the Community Earth System Model Version 2 Large Ensemble (CESM2-LE)[51] is used to elucidate the impact of external forcing on decadal variability in the North Atlantic SST. Unlike the Coupled Model Intercomparison Project (CMIP), which contains uncertainties arising from model structure and physics, a single model initial-condition large ensemble such as the CESM2-LE contains a set of simulations from a single climate model under an identical forcing scenario[52]. The ensemble spread thus arises solely from perturbations in the initial conditions. This approach facilitates the separation of the forced component (ensemble mean) and internal variability (deviations from ensemble mean), enabling the assessment of changes in internal variability in response to external forcing. In this study, we identify pronounced differences in the trajectories of northern North Atlantic SST among ensemble members during the mid-twenty-first century. We attribute these distinct trajectories to the different rates of North Atlantic deep convection reduction among ensemble members. We hypothesize that these differences are initiated by stochastic atmospheric variability and amplified by positive feedbacks activated by the mean freshwater forcing induced by global warming.

## Results

### Change of internal decadal variability over time

Decadal variability in North Atlantic SST is isolated by applying a lowpass filter (11-year running mean) to winter (December-January-February-March, DJFM) North Atlantic SST for each member of the CESM2-LE. We find an increased ensemble spread across the 100 ensemble members during the mid-twenty-first century that is concentrated north of 50°N (see Supplementary Movie 1), indicating a wider range of possible future northern North Atlantic climate states compared to those in the historical period. This region is among the regions with the highest predictability in initialized decadal predictions under current climate conditions[53–56]. To evaluate changes in this region under global warming, an index of northern North Atlantic SST (NNASST) is computed as the area-average over the 50–80°N, 90°W–40°E domain (Supplementary Fig. 1a). This analysis identifies a pronounced increase in NNASST ensemble standard deviation from 2040–2070, with the maximum standard deviation occurring during 2056–2060 (gray line in Fig. 1a).

This increase in North Atlantic internal variability might be stochastic in nature and thus unpredictable, or it could stem from processes that, when traced back, may provide some predictability. To identify the onset of the broadened SST distribution and any potential precursors, we classify ensemble members into warm and cold groups, then conduct composite analyses of these groups. The classification of ensemble members is based on their winter NNASST during the 5 years with the highest NNASST standard deviation (2056–2060). Specifically, ensemble members are assigned to the warm (cold) group if their NNASST index is greater (smaller) than or equal to the ensemble mean plus (minus) one standard deviation of the NNASST in the CESM2 preindustrial control (piControl) simulation[57] during any of these 5 years. The classification yields 40 members in the warm group and 31 members in the cold group and is not sensitive to the choice of season (DJF vs. DJFM vs. annual) or domain. Our results are also robust to different classification methods (Supplementary Figs. 2, 3).

Figure 1a displays the ensemble mean and spread of the lowpass-filtered DJFM NNASST index for both the warm and cold groups from 1920 to 2094. The ensemble means of these two groups diverge significantly from one another, starting from 2035 and persisting until the end of the simulation, with the maximum separation occurring during 2050–2070. This result implies that there are distinct trajectories of NNASST in these groups that start ~20 years before the maximum spread in internal variability. Furthermore, if it were possible to identify which trajectory is being taken, it would suggest the potential for multidecadal predictability of NNASST in the mid-twenty-first century.

The distinction between the cold and warm groups emerges from the Labrador Sea region in the 2030s and expands progressively across the entire mid-to-high latitude North Atlantic in the following decades, indicating an earlier formation of the North Atlantic warming hole in the cold group (Fig. 2). Based on this spatial development, we attribute part of the large-scale SST difference between the groups to the transport of colder water from the Labrador Sea to the subpolar North Atlantic in the cold group compared to the warm group (see the following section for more details). In addition to the transport from the Labrador Sea, the AMOC plays a dominant role in poleward heat transport[40], which can be tied back to deep convection in the Labrador Sea in CESM[20,35,58,59]. As such, we assess the Labrador Sea mixed layer depth (MLD), northward heat transport (NHT) across 50°N, and the AMOC at 50°N for both the warm and cold groups (Fig. 1b–d) to investigate their potential roles in driving the distinct NNASST trajectories among ensemble members.

All ensemble members exhibit continuously shoaling Labrador Sea MLD (Fig. 1b), consistent with increased ocean stratification due to anthropogenic forcing[37]. In the mid-twenty-first century, the Labrador Sea deep convection in all ensemble members gradually stops, with final MLDs fluctuating around 80 m. However, a clear distinction between the warm and cold groups emerges from 2029. The cold group has shallower MLDs, with an accelerated reduction of the Labrador Sea deep convection relative to the warm group. This acceleration culminates in an earlier Labrador Sea deep convection shutdown in the cold group, as can be seen in the 11-year difference in the median time at which MLD reaches 80 m: 2052 for the cold group and 2063 for the warm group (Fig. 1b and Supplementary Fig. 4).

The difference in the deep convection between the warm and cold groups leads to a clear separation in their AMOC and NHT trajectories starting 2032 and 2029, respectively (Fig. 1c, d). The cold group has slightly weaker AMOC, which leads to reduced NHT compared to the warm group. In addition to AMOC, NHT is also impacted by changes in gyre circulation, which is known to respond more rapidly to stochastic atmospheric forcing than AMOC[35,60]. This gyre response might explain why the separation of NHT trajectories occurs a few years earlier than the separation of AMOC between the groups. These differences in NHT would act to intensify the differences in NNASST trajectories (Fig. 1a).

### Processes driving distinct density, salinity, and temperature in the upper Labrador Sea

Given that the difference in MLD shutdown rates between the groups precedes the differences in AMOC, NHT, and NNASST, it is crucial to investigate what causes the MLD difference between the groups. To start with, we examine the time evolution of the upper-1500 m ocean density averaged over the Labrador Sea region (Supplementary Fig. 1b) and display the difference between the cold and warm groups in Fig. 3a. Consistent with the MLD difference between the groups, the upper-ocean density in the cold group becomes significantly smaller than that in the warm group from the 2020s, with the maximum difference reached in the 2050s and located in the upper 100 m.

The density of seawater depends primarily on temperature and salinity. Thus, we assess the contributions of temperature ($-\alpha_\theta \Delta \theta$) and salinity ($\beta_S \Delta S$) to the differences between the groups in the upper 1500 m over time (Fig. 3b, c). Here, $\alpha_\theta$ and $\beta_S$ represent the thermal

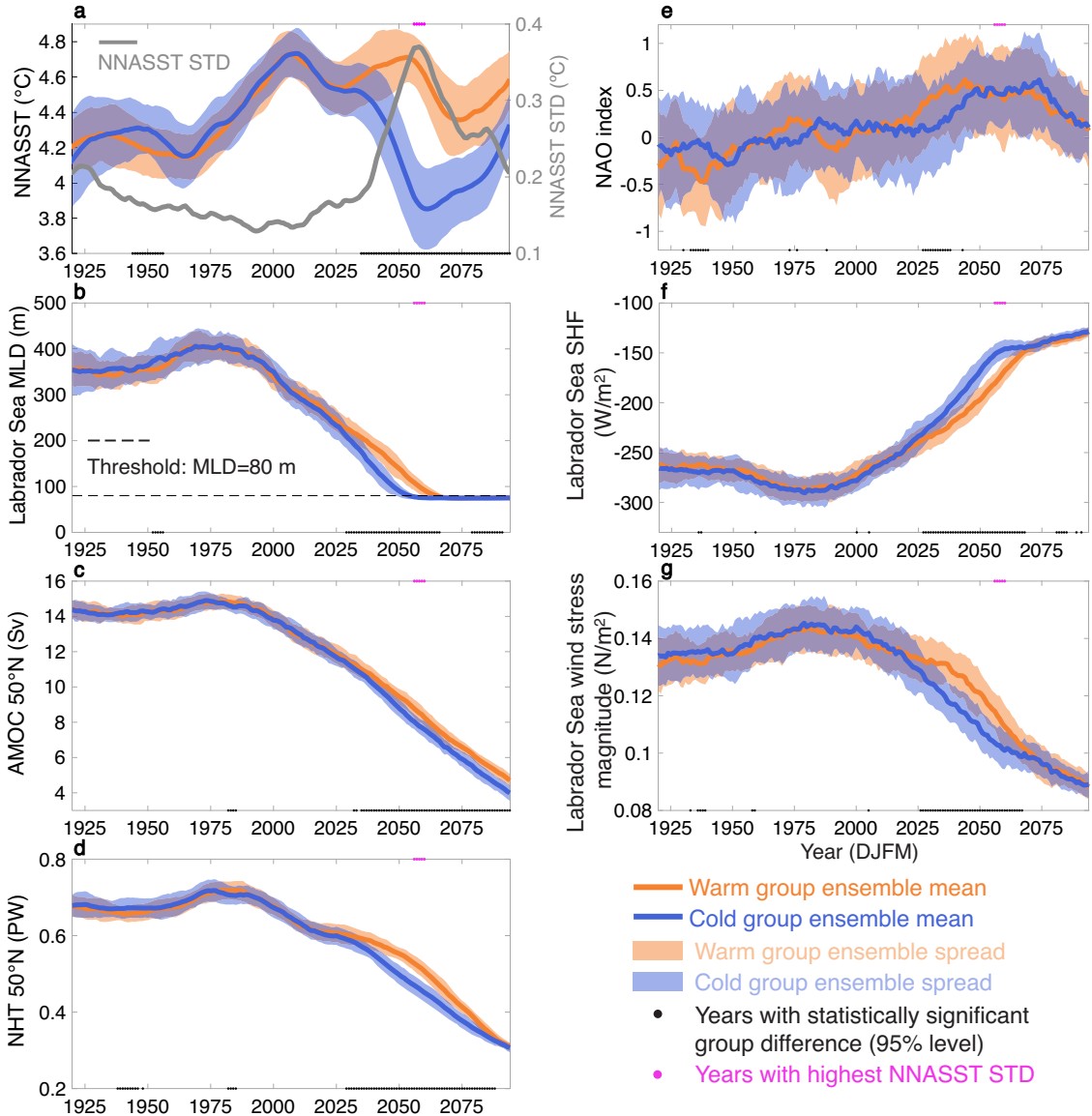

**Fig. 1 | Lowpass-filtered December-January-February-March (DJFM) time series averaged in the warm (orange) and cold (blue) groups. a** Area-weighted average sea surface temperature (SST) in the North Atlantic, north of 50°N (NNASST; domain shown in Supplementary Fig. 1a; °C). **b** Area-weighted average mixed layer depth (MLD) in the Labrador Sea (domain shown in Supplementary Fig. 1b; m). **c** 50°N Atlantic meridional overturning circulation (AMOC) index (Sv) defined as the maximum overturning streamfunction (Eulerian-mean component) below 500 m at 50°N. **d** Northward heat transport (NHT) at 50°N (PW), with this latitude selected based on the NNASST domain. **e** North Atlantic Oscillation (NAO) index defined as the difference in normalized DJFM sea level pressure between the model grid points closest to Lisbon, Portugal and Stykkisholmur, Iceland. **f** Labrador Sea

surface heat flux (SHF; W/m²), with negative values indicating heat loss from the ocean to the atmosphere. **g** Magnitude of the Labrador Sea wind stress (N/m²). The shadings are the ensemble spread, calculated as ensemble mean ± one ensemble standard deviation (STD), for the warm (orange) and cold (blue) groups. Black dots on the lower axis signify years when these ensemble means are significantly different at the 95% level based on a two-tailed Student's *t*-test. The same significance test applies to Figs. 2–4, 6. Magenta dots on the upper axis mark years 2056–2060, which are the 5 years with the maximum STD of NNASST. The gray line in (**a**) is the STD of NNASST across 100 ensemble members. The dashed black line in (**b**) is a MLD threshold of 80 m, which is used to generate the histogram in Supplementary Fig. 4.

expansion coefficient and haline contraction coefficient, respectively, and $\theta$ and $S$ denote potential temperature and salinity (see Methods for details). Notably, the pattern in Fig. 3c closely resembles that in Fig. 3a, whereas Fig. 3b shows a reduced magnitude with an opposite sign. This suggests that salinity primarily drives the density difference between the groups, while the temperature has a minor damping effect. That is, the cold group has smaller upper-ocean density mainly because it is fresher than the warm group.

In light of the dominant contribution of salinity, we conduct a salinity budget analysis over the upper 295 m in the Labrador Sea region (see Methods for details). The differences between the cold and

warm groups for each term are depicted in Fig. 4a. In the Labrador Sea region, a significantly smaller salinity tendency is seen in the cold group compared to the warm group during winter from the 2020s to 2058 (Fig. 4a). The primary contributor to the smaller total salinity tendency in the cold group before 2060 is the difference in diabatic vertical mixing between the groups, which begins in the 2020s and peaks in the 2050s. The impact of vertical mixing on the upper Labrador Sea salinity variability has also been shown in previous studies[61,62]. Contributions from the remaining four terms are smaller but, at times, still significant. Surface flux counteracts the total salinity tendency difference, primarily due to reduced melting and

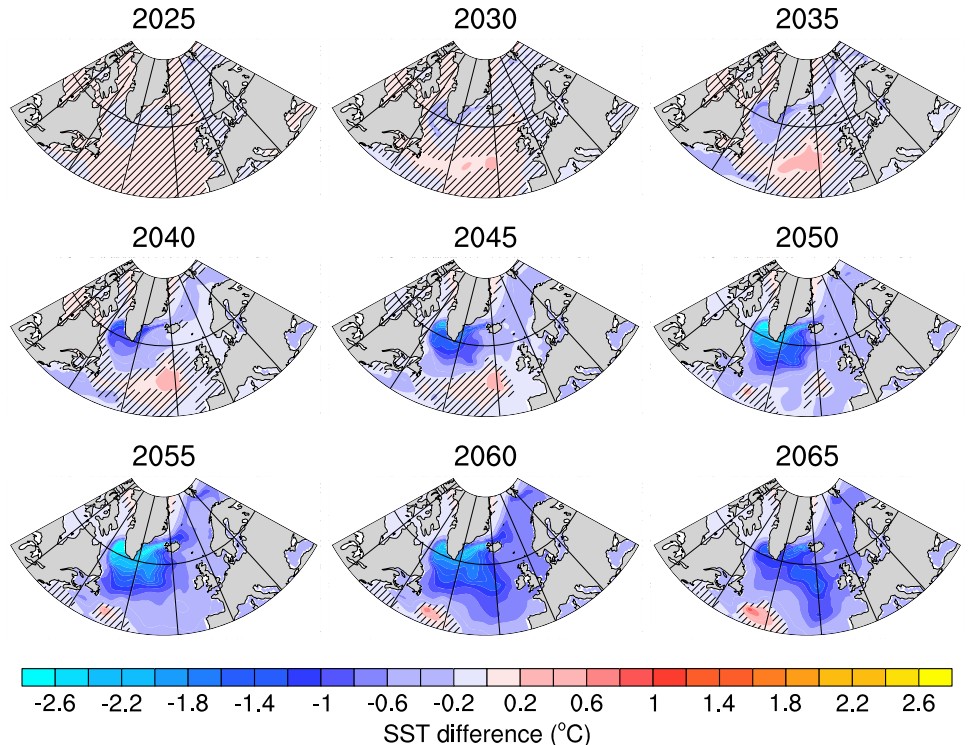

**Fig. 2 | Spatial evolution of sea surface temperature (SST) difference between the groups.** Composite of lowpass-filtered December-January-February-March SST (°C) differences between the cold and warm groups (cold group–warm group) for every 5 years from 2025 to 2065. The absence of black hatching indicates significant differences at the 95% level.

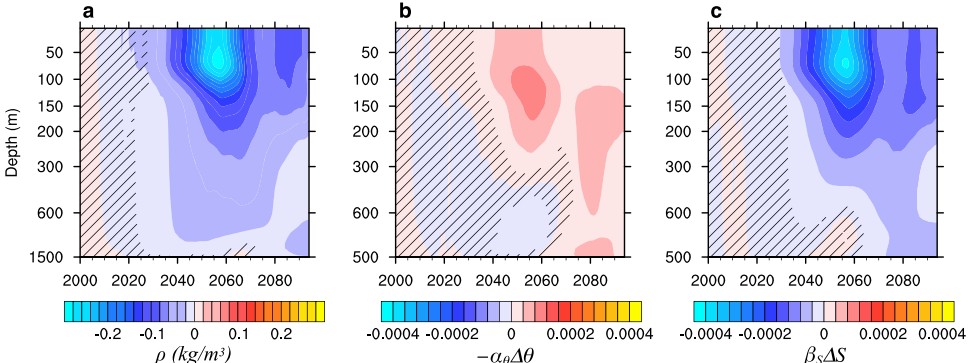

**Fig. 3 | Decomposition of density for the groups. a** Differences between the cold and warm groups (cold group–warm group) in the lowpass-filtered December-January-February-March Labrador Sea density ($\rho$, kg/m³). **b** Same as (**a**) but for temperature's contribution to the density ($-\alpha_\theta\Delta\theta$). **c** Same as (**a**) but for salinity's contribution to density ($\beta_S\Delta S$). Here, $\alpha_\theta$ and $\beta_S$ are the thermal expansion coefficient and haline contraction coefficient, respectively, and $\theta$ and $S$ are potential temperature and salinity (see Methods for details). The absence of black hatching indicates significant differences at the 95% level.

precipitation in the cold group relative to the warm group (Supplementary Fig. 5a). Parameterized advection has a minor opposing effect until about 2075. Resolved advection amplifies the negative total tendency during 2031–2040, followed by an opposing contribution from 2049. This transition in resolved advection, along with sustained opposing contributions from surface flux and parameterized advection, is responsible for the larger total salinity tendency in the cold group from the 2060s as the influence of vertical mixing declines. Lastly, the residual term modestly reinforces the negative total tendency beginning in 2036. This term encompasses lateral diffusion and K-Profile vertical mixing Parameterization (KPP;[63]) non-local vertical mixing that are not available in the simulation employed in this study, as well as a small term, the Robert Filter tendency.

This analysis implies that the fresher upper Labrador Sea in the cold group primarily results from weaker vertical mixing compared to the warm group, limiting the upward transport of higher-salinity sub-surface seawater. This leads to weaker deep convection in the cold group, which can, in turn, influence its vertical mixing (see the following section for details). These processes reduce the NNASST in the cold group through changes in AMOC. Another pathway through which the two groups diverge in their trajectories is via seawater transport from the Labrador Sea to the subpolar North Atlantic (Fig. 2), as discussed in the previous section. Therefore, we also conduct a heat budget analysis in the upper-295 m Labrador Sea for both groups, even though the temperature has only a minor damping effect on the Labrador Sea density difference between the groups (Fig. 3).

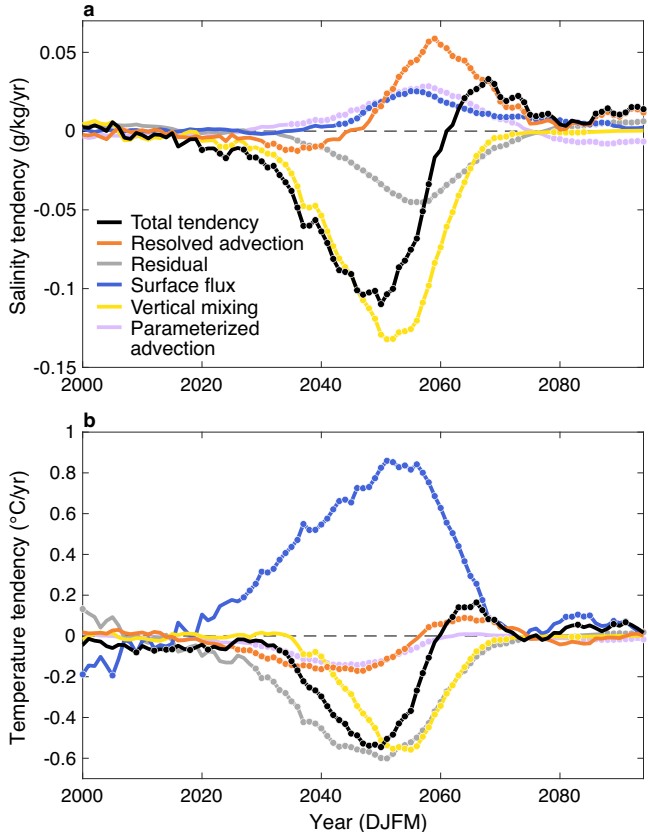

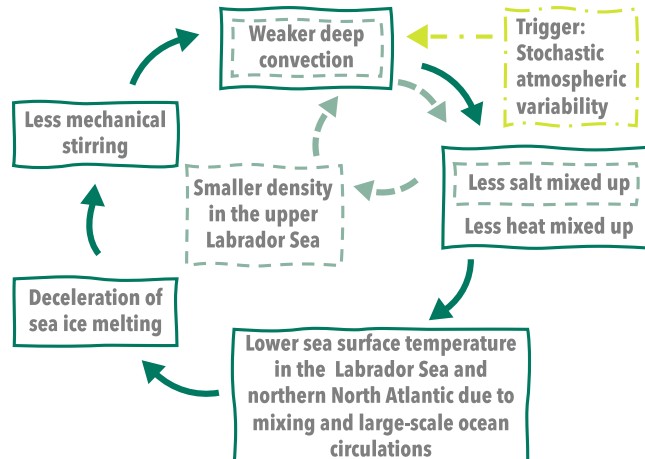

**Fig. 5 | Schematic of the trigger and two proposed positive feedbacks.** The trigger is represented by the dash-dotted yellow-green box and arrow. The vertical mixing-surface salinity feedback is represented by dashed light-green boxes and arrows, and the sea ice-wind stress feedback is represented by solid dark-green boxes and arrows.

**Fig. 4 | Salinity and heat budget analysis in the upper 295 m of the Labrador Sea. a** Differences between the cold and warm groups (cold–warm) in total salinity tendency (g/kg/yr; black), as well as contributions to the total tendency from resolved advection (orange), diabatic vertical mixing (yellow), surface flux (blue), parameterized advection (purple), and residual terms (gray). Dots indicate the years when the warm and cold groups are significantly different at the 95% level. **b** Same as (**a**) but for the temperature tendency (°C/yr). All time series are December-January-February-March-averaged and lowpass-filtered before calculating the differences.

The cold group has a significantly smaller total temperature tendency than the warm group from the early 2030s to the late 2050s (Fig. 4b). One of the largest contributions to their difference is from vertical mixing. Consistent with the salinity budget analysis (Fig. 4a), this indicates that the cold group has weaker vertical mixing compared to the warm group, which results in reduced upward transport of warm subsurface seawater. The vertical mixing term is important within regions characterized by active deep convection, such as the Labrador Sea (Supplementary Fig. 6). This may explain why the Labrador Sea emerges as a key area for the large-scale SST difference between the groups (Fig. 2). Alongside vertical mixing, the residual term is also a primary contributor to the group difference. Similar to the salinity budget analysis, this term mainly consists of lateral diffusion and KPP non-local vertical mixing that are not available in CESM2-LE. Additionally, both resolved and parameterized advection moderately contribute to the group difference. In contrast, surface heat flux consistently dampens the difference between the groups starting from around 2025, which can be primarily attributed to the reduced turbulent heat flux from the ocean to the atmosphere in the cold group (Supplementary Fig. 5b).

### Triggers, feedbacks, and timing of the divergent states among ensemble members

A missing element in the discussion above is the initial source of the difference between the warm and cold groups. Figure 1e shows the lowpass-filtered DJFM North Atlantic Oscillation (NAO) index for the warm and cold groups. The cold group has significantly smaller NAO index than the warm group from 2027 to 2038. This variability can lead to either decreased heat loss from the Labrador Sea on short time-scales through associated reduced westerlies (Fig. 1f), weaker mechanical stirring through wind stress that results in reduced vertical salinity mixing (Figs. 1g, 4a), or both within the cold group. These mechanisms are hypothesized to trigger the initial difference in the Labrador Sea deep convection between the groups.

This difference in stochastic NAO variability is a common occurrence within the period of study, and it only persists for a short time period. Specifically, a comparable difference in the NAO index is evident around 1935 between the two groups, even with the current classification (Fig. 1e). This NAO difference, spanning approximately a decade, leads to a significant difference in NNASST for multiple years around 1950 between the groups (Fig. 1a). However, this difference in NNASST is relatively short-lived and does not intensify, in contrast to the divergence starting from 2035. As such, there must be amplification processes that facilitate the intensification and persistence of the divergent climate states among ensemble members. Here, we propose two feedback mechanisms that may be amplifying and extending the impact of the stochastic trigger, maintaining the divergent trajectories of NNASST.

The first positive feedback is associated with the close relationship between Labrador Sea deep convection and vertical mixing. That is, in the cold group, the initially weaker deep convection restricts the upward mixing of higher-salinity subsurface seawater, which further weakens deep convection by reducing the density in the upper Labrador Sea. This feedback can accelerate the rate of deep convection shutdown in the cold group compared to the warm group (light-green; Fig. 5). The dominant contribution of vertical mixing to the salinity tendency difference between the groups (Fig. 4a), and the dominant contribution of salinity to the density difference (Fig. 3) are evidence that this positive feedback is occurring.

The second positive feedback mechanism involves the interactions between sea ice, wind stress, and vertical mixing (dark-green; Fig. 5). In the cold group, weaker deep convection within the Labrador region causes the Labrador Sea to become relatively colder, due to reduced vertical mixing. Simultaneously, the entire North Atlantic basin cools relative to the warm group because of large-scale ocean circulation (e.g., AMOC, Fig. 1c) and transport originating from the

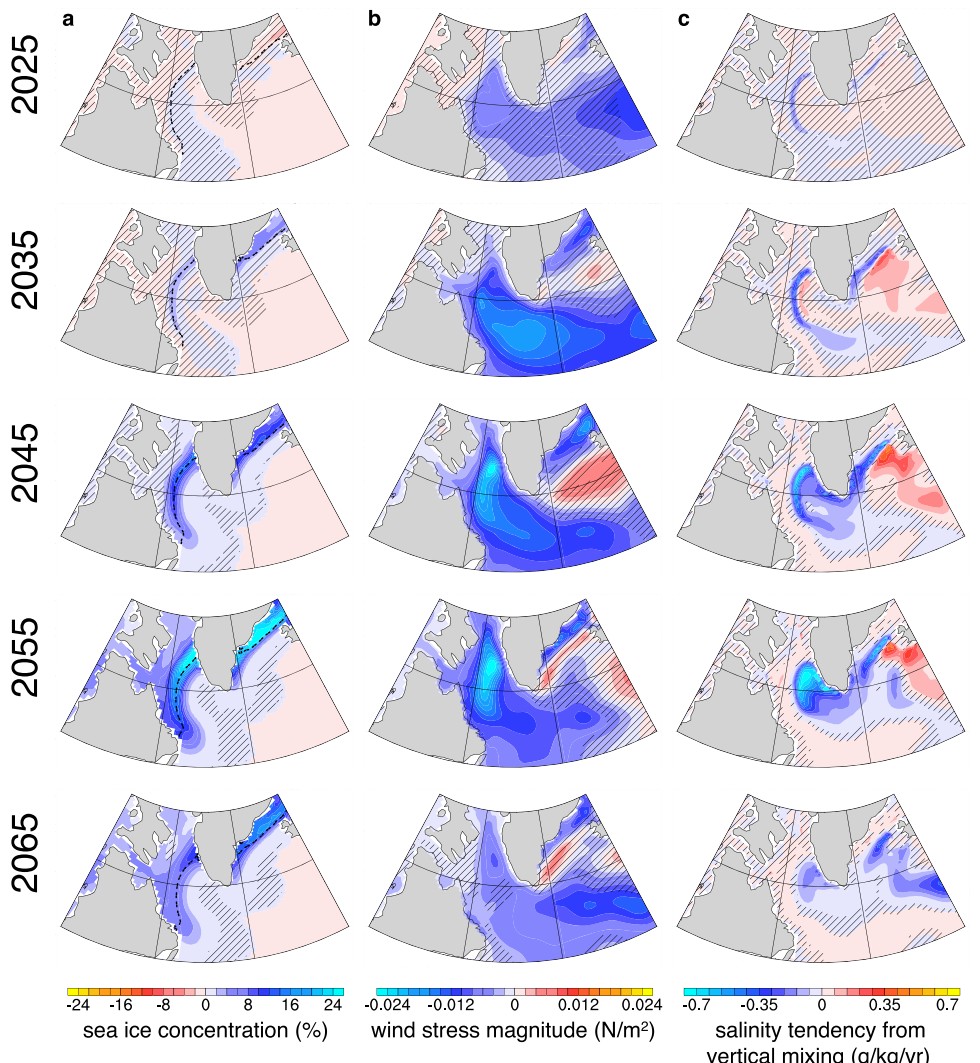

**Fig. 6 | Evidence for the sea ice-wind stress positive feedback.** Lowpass-filtered December-January-February-March differences between the cold and warm groups (cold group–warm group; shadings) every 5 years from 2025 to 2065 for **a** sea ice concentration (%), **b** wind stress magnitude (N/m²), and **c** contribution to salinity tendency from diabatic vertical mixing for the upper 295 m (g/kg/yr). The absence of black hatching indicates significant differences at the 95% level. In **a**, dashed black line denotes the 100-member ensemble mean sea ice edge defined as the 15% ice concentration contour for the corresponding year.

Labrador Sea (Fig. 2). These processes contribute to reduced sea ice melting in the cold group compared to the warm group (Fig. 6a), and the positive ice-albedo feedback[64] may further amplify the differences in SST and sea ice between the groups. The reduction in sea ice melting makes the surface ocean in the cold group less susceptible to wind stress, particularly along the sea ice edge (Fig. 6b). Consequently, the cold group experiences less mechanical stirring than the warm group, leading to less salt being mixed upward (Fig. 6c). This can further weaken deep convection and result in colder SST in the cold group. Evidence for this proposed mechanism can be found in the group differences in sea ice cover (Fig. 6a), wind stress magnitude (Fig. 6b), and salinity tendency due to vertical mixing (Fig. 6c), all of which occur near the sea ice edge. It is notable that anomalies in Fig. 6c do not extend as far north, confined by their climatology (Supplementary Fig. 7). This positive feedback explains the long-lasting difference in the wind stress magnitude between the warm and cold groups, which remains significant even when the NAO index no longer displays a significant difference between them (Figs. 1e, g, 6b).

The identification of these feedbacks does not explain why the increase in internal variability occurs at this specific time. Figure 7 shows the ensemble mean salinity and its vertical gradient in the Labrador Sea. The vertical salinity gradient increases over time due to the continuous freshening of the surface ocean, which can be attributed to Arctic sea ice melting[37,45] and changes in Arctic runoff[65] caused by global warming. As a result, vertical mixing acting on this enhanced gradient can lead to greater modifications of the surface salinity.

Consequently, the two positive feedback mechanisms, which are heavily reliant on surface salinity fluctuations owing to vertical mixing, become particularly active from the early twenty-first century when the vertical salinity gradient increases. This indicates that external forcing drives the divergent trajectories of NNASST among ensemble members (i.e., a wider range of possible climate states due to internal variability). Similarly, a re-convergence of these trajectories can also be due to external forcing, as surface freshening can cause the cessation of deep convection in the Labrador Sea across all ensemble members[37,45], ultimately leading to the termination of the related feedback mechanisms.

### Changes in internal SST variability in other large ensembles
As this study focuses on CESM2-LE, the physical processes revealed may depend on certain specifics of this model, such as the dominant location of deep water formation (Labrador Sea) and parameterization

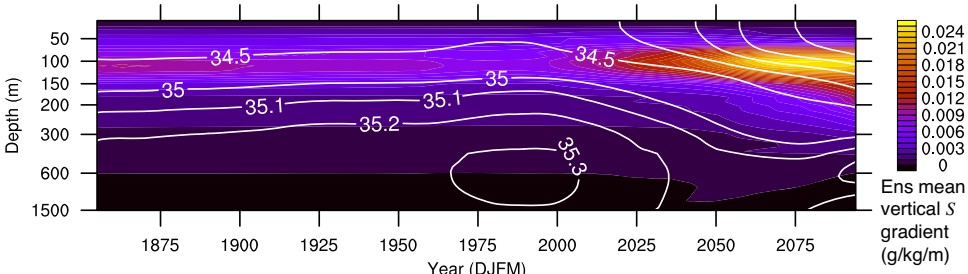

**Fig. 7 | Ensemble mean salinity and its vertical gradient.** Lowpass-filtered December-January-February-March ensemble mean salinity (contours) and its vertical gradient (shading; approximated as the difference in salinity divided by the difference in depth between two adjacent layers) averaged over the Labrador Sea region across 100 ensemble members.

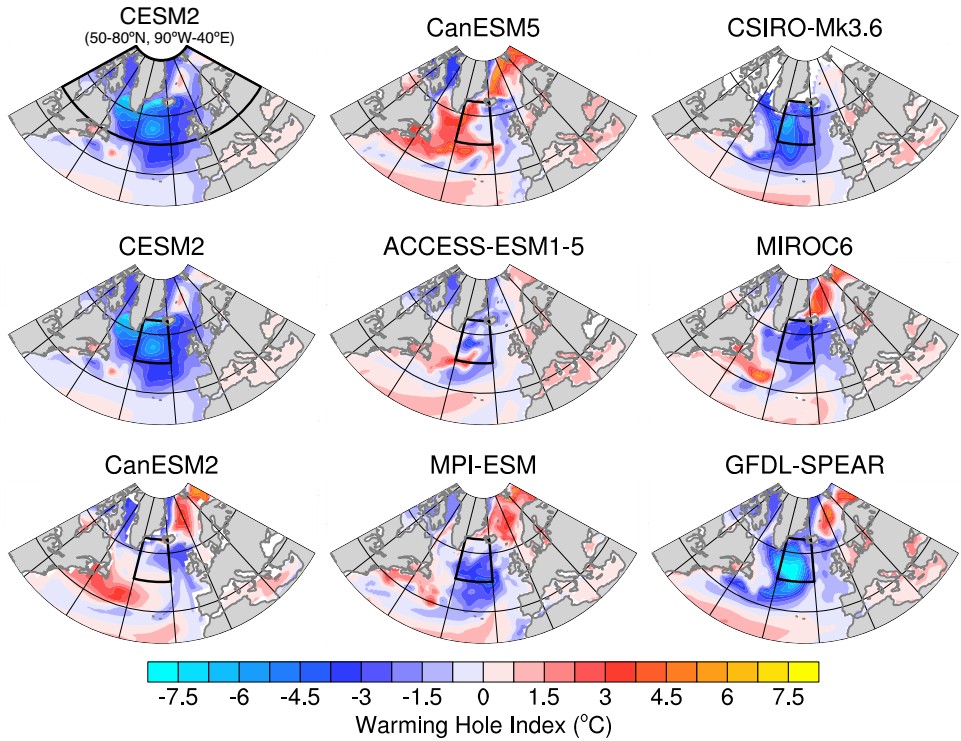

**Fig. 8 | The response of the North Atlantic sea surface temperature (SST) to global warming in multiple large ensembles.** The shading in each subplot shows the ensemble mean warming hole index (°C; see Methods for details) at each grid cell for each large ensemble. The black box denotes the domain used to calculate the area-weighted average SST standard deviation across all members of each large ensemble in Fig. 9.

schemes. To evaluate the generalizability of our findings, we utilize seven additional large ensembles to assess whether an increased ensemble spread of NNASST exists in other models during the mid-twenty-first century. These large ensembles include models from both CMIP5 and CMIP6, under the historical and one of three future radiative forcing scenarios, namely the Shared Socioeconomic Pathway (SSP) 3-7.0, SSP 5-8.5, and Representative Concentration Pathway (RCP) 8.5 (see Methods for details).

In the CESM2-LE, we identified a discrepancy of SST among different ensemble members primarily in the Labrador Sea and subpolar North Atlantic and extending further to the Greenland-Iceland-Norwegian (GIN) Seas (Fig. 2 and Supplementary Movie 1). Based on the mechanisms identified in our study, the spatial distribution of SST discrepancy is linked to the mean response of deep convection, AMOC, and North Atlantic SST (i.e., the development of a warming hole) to global warming. These responses vary across different models, as documented in previous studies[50,66]. Therefore, before delving into the impact of external forcing on internal SST variability in other

models, we first examine the ensemble-mean North Atlantic response to global warming. This assessment is conducted using an SST warming hole index that is shown to be closely associated with MLD and AMOC[66], as the SST output is widely available in all the models (see Methods for details). Figure 8 reveals that a considerable number of models simulate cooling in the central subpolar North Atlantic in future climates compared to the future global mean SST, indicating the emergence of a North Atlantic warming hole. The location and magnitude of the warming hole vary among models, consistent with ref. 66. Some models (MPI-ESM, MIROC6, GFDL-SPEAR) have a robust warming hole within the subpolar North Atlantic and opposite signed anomalies in the GIN Seas. Additionally, certain models (CanESM2, CanESM5, and ACCESS-ESM1-5) have weaker cooling and/or mixed warming and cooling signals across the subpolar North Atlantic that are not consistent with the development of a North Atlantic warming hole.

Due to the inter-model spread in the warming hole location and the close relationship between the formation of the warming hole and

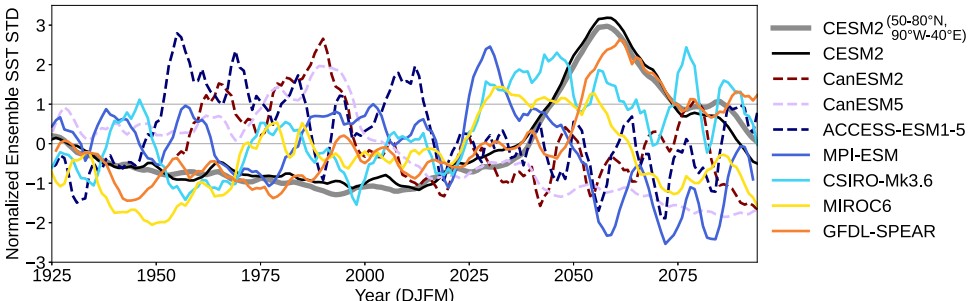

**Fig. 9 | North Atlantic sea surface temperature (SST) ensemble spread for multiple large ensembles.** Each line represents the standard deviation of the lowpass-filtered, area-averaged December-January-February-March SST across all the members within each large ensemble, normalized along the time dimension for each model. The domain used for the area-weighted average is shown in Fig. 8. The gray line is the normalized version of the one shown in Fig. 1a, serving as a reference.

the increase of ensemble spread (e.g., Fig. 1a), we use a smaller common domain in the central subpolar North Atlantic (50-65°N, 40-20°W) that captures the North Atlantic warming hole across the majority of models to measure the SST ensemble spread over time for all large ensembles. Figure 9 displays the ensemble spread of the lowpass-filtered DJFM SST across all the members within each large ensemble, normalized along the time dimension for each model. The ensemble spread of CESM2-LE in the original broad domain (50-80°N, 90°W-40°E) and in the small domain remains similar. All the models with a warming hole (CESM2, MPI-ESM, CSIRO-Mk3.6, MIROC6, and GFDL-SPEAR) show a distinct increase in SST ensemble spread in the twenty-first century, albeit with different timing (Fig. 9). This variation in timing is expected, as the models differ in their physics and some are subject to different forcings. On the contrary, the three models without warming hole formation (CanESM2, CanESM5, ACCESS-ESM1-5) exhibit a decrease in the ensemble spread in the twenty-first century compared to the 20th century. It is noteworthy that limiting the domain for these "non-warming hole models" to only the relative cooling region still does not result in an increase of ensemble spread in the mid-twenty-first century. These results suggest that if a model experiences distinct deep convection shutdown and AMOC slowdown, which manifests as the formation of a North Atlantic warming hole (Fig. 8), it will experience an increase in internal variability in the subpolar North Atlantic (Fig. 9). The consistent behavior seen across all the "warming hole models" underscores the robustness of our results.

## Discussion

Previous studies have offered valuable insights into the mechanisms driving North Atlantic SST variability using observations, piControl simulations, and historical simulations with limited realizations[29-35]. In this study, we utilize a single model initial-condition large ensemble containing 100 simulations that can allow us to study internal variability within the context of changes in external forcing. This addresses a gap left by previous studies, enabling us to diagnose how anthropogenic external forcing can modulate internal variability and how their interactions may improve predictability on decadal timescales.

In this study, we uncover how future climate change might impact decadal variability in the North Atlantic Ocean. Using the CESM2-LE simulations, we find that the range of potential climate states in the northern North Atlantic increases dramatically in the mid-twenty-first century. We propose that distinct trajectories of NNASST originate from stochastic atmospheric variability. As global warming shifts the mean ocean state, it eventually crosses a critical "tipping point", activating positive feedbacks. These feedbacks introduce pronounced nonlinearity into the system, magnifying the initial stochastic differences. Our findings thus highlight the critical role of external forcing in driving increased internal variability.

Additional investigation of seven other large ensembles shows that this increase in internal SST variability broadly exists in models that simulate the formation of a North Atlantic warming hole under global warming, despite variations in timing across different models. While there exist debates, recent studies have provided evidence for the slowdown of the AMOC and the formation of the North Atlantic warming hole in the observations[44,67-69]. This evidence underscores the potential real-world manifestation of our findings, warranting further attention from our community.

In the CESM2-LE, we identify distinct trajectories starting around 2030 that lead to either warm or cold NNASST, which then lasts for decades until the end of the simulations. This suggests a potential increase in the multidecadal predictability of North Atlantic SST, despite increased internal variability. Such added predictability on decadal timescales over the upcoming 50−70 years could hold significant socioeconomic value, informing climate adaptation and mitigation strategies. Here, we propose two potential avenues for future work that can leverage this enhanced predictability in reality.

The first avenue involves the comparison between the observations and the model simulations. The distinct trajectories between the warm and cold groups in CESM2-LE can be identified in multiple variables (e.g., NNASST, MLD, NHT, AMOC) starting from ∼2030−only a few years from now. However, with the exception of SST, the majority of these do not have sufficiently long records in the observations. Monitoring the NNASST in the observations over the upcoming decade and comparing this to the envelope of NNASST in the CESM2-LE warm and cold groups may allow us to infer whether the observations are consistently within one trajectory or the other. From this, we may infer the North Atlantic climate state in the following 50−70 years.

The second avenue for future work is related to dynamical model predictions. Current decadal prediction models generally provide predictions out to about 10 years[55,56]. The extended predictability revealed here suggests that these models could potentially offer skillful predictions over longer periods once the mean climate allows the activation of positive feedbacks. The differing timing of the increase in subpolar SST standard deviation suggests that the timing of crossing the "tipping point" can vary across different models. Our analysis with CESM2-LE indicates that the divergence in trajectories is triggered around 2027 in this model, pinpointing this year as the approximate time when the mean state crosses the "tipping point" that activates the positive feedbacks. Thus, we may extend the prediction period beyond the typical 10 years for dynamical predictions initialized around 2027 in the CESM2 to achieve skillful long-lead predictions.

In addition to the anthropogenic external forcing and internal variability focused on in this study, previous studies have shown that natural external forcing, such as volcanic eruptions can facilitate skillful decadal SST prediction in the North Atlantic through dynamical

mechanisms involving NAO and AMOC[70–72]. Thus, another avenue for future work is to explore how the climate impacts and predictability associated with natural external forcing may change under global warming.

## Methods

### Model simulations: CESM2

This study uses the CESM2[57] 100-member Large Ensemble simulations[51]. CESM2 is a fully coupled Earth system model with the Community Atmosphere Model version 6 (CAM6), the Parallel Ocean Program version 2 (POP2), the Community Ice CodE Model version 5 (CICE5), and the Community Land Model version 5 (CLM5) as its components. All components have a nominal 1° horizontal resolution. Compared to CESM1, the enhanced model physics in CESM2 results in notable improvements including reductions in tropical precipitation biases, more realistic representation of teleconnection patterns, and better freshwater exchange in the estuaries, all of which improve CESM2's representation as compared to observations[57]. Of particular relevance to this study, CESM2 can accurately capture climate variability on decadal timescales[73,74].

CESM2-LE is forced with the CMIP6 historical radiative forcing scenario (1850–2014) and the SSP 3-7.0 future radiative forcing scenario (2015–2100), which is a medium-to-high emissions scenario[75]. Ensemble spread is generated using a combination of various oceanic and atmospheric initial states[51]. Among the 100 ensemble members, 80 were initialized from four pre-selected years of a CESM2 piControl simulation[57], each representing a different AMOC phase. There are 20 members initialized from each of the four AMOC states with ensemble spread created by roundoff-level perturbations in the initial atmospheric potential temperature field, referred to as micro-perturbations. Another 20 members were initialized 10 years apart between years 1001 and 1191 of the piControl simulation to incorporate additional, so-called, macro-perturbations. Furthermore, half of these 100 members were forced with the CMIP6 biomass burning (BMB) emissions during 1997–2014, which contain larger interannual variability compared to the data sources utilized before and after this period[76,77]. In contrast, the other half used a lowpass-filtered (11-year running mean) version of the CMIP6 BMB data during this period, which reduced the discontinuity in BMB variability.

In addition to the CESM2-LE, we use a 1601-yr piControl simulation from CESM2 (model years 0400–2000) to characterize the internal variability in the North Atlantic, with the initial 399 years excluded from the analysis to account for model spin-up. This long simulation, devoid of external forcings, permits a thorough investigation of the full range of internal variability in the model.

In this study, we focus on winter variability, as the reemergence of SST anomalies during winter leads to longer memory and increased predictability compared to other seasons[40,78–82]. In addition, the NAO variability and thus its associated impacts on the ocean are the most prominent during wintertime[34]. Finally, an extended winter season, including March, is examined because the climatological MLD in the Labrador Sea reaches its peak during this month. Therefore, variables are subject to a DJFM seasonal mean and a subsequent 11-year running mean lowpass filtering to isolate decadal variability.

### Model simulations: additional large ensembles

In addition to the CESM2-LE, we use seven large ensembles, including both CMIP5 and CMIP6 models. These large ensembles contain at least 25 members. The simulation time period, forcing, and ensemble size of each large ensemble used here are listed in Table 1.

### Labrador Sea density decomposition

In this study, we assess the contributions of temperature and salinity to the Labrador Sea density based on a linear equation of state[83] of the

**Table 1 | The model (reference), time period, forcing, and ensemble size of the single model initial-condition large ensembles used in this study**

| Model | Time period | Forcing | Ensemble size |
|---|---|---|---|
| CESM2[51] | 1850–2100 | Historical, SSP 3-7.0 | 100 |
| CanESM2[88] | 1950–2100 | Historical, RCP 8.5 | 50 |
| CanESM5[89] | 1850–2100 | Historical, SSP 3-7.0 | 25 |
| ACCESS-ESM1-5[90] | 1850–2100 | Historical, SSP 3-7.0 | 40 |
| MPI-ESM[91] | 1850–2099 | Historical, RCP 8.5 | 99 |
| CSIRO-Mk3.6[92] | 1850–2100 | Historical, RCP 8.5 | 30 |
| MIROC6[93] | 1850–2100 | Historical, SSP 3-7.0 | 50 |
| GFDL-SPEAR[94] | 1921–2100 | Historical, SSP 5-8.5 | 30 |

form:

$$\rho = \rho_0[1 - \alpha_\theta(\theta - \theta_0) + \beta_S(S - S_0)], \quad (1)$$

where $\alpha_\theta$ is the thermal expansion coefficient defined as

$$\alpha_\theta = -\frac{1}{\rho}\frac{\partial \rho}{\partial \theta}, \quad (2)$$

and $\beta_S$ is the haline contraction coefficient defined as

$$\beta_S = -\frac{1}{\rho}\frac{\partial \rho}{\partial S}. \quad (3)$$

$\rho$, $\theta$, and $S$ are the density, potential temperature, and salinity, and the reference values $\rho_0$, $\theta_0$, and $S_0$ are derived as the DJFM mean Labrador Sea upper-295 m volume-average of their corresponding variables in the CESM2 piControl simulation. The Labrador Sea domain is defined using POP2's default ocean region masks, with a northern boundary set at 65°N (Supplementary Fig. 1b). By comparing the magnitudes of $-\alpha_\theta\Delta\theta$ and $\beta_S\Delta S$ where $\Delta\theta$ equals $\theta - \theta_0$ and $\Delta S$ equals $S - S_0$, we can determine the relative contributions of temperature and salinity to changes in density.

### Heat and salinity budget analysis

To develop a mechanistic understanding of the processes influencing the temperature and salinity in the upper Labrador Sea, we perform ocean heat and salinity budget analyses. In the heat budget analysis, we decompose the total temperature tendency in the upper 295 m following refs. [84] and [85]:

$$\frac{1}{H}\int_{-D}^{\eta}\frac{\partial\theta}{\partial t}dz = \frac{1}{H}\int_{-D}^{\eta}(-\nabla\cdot\mathbf{u}\theta)dz + \frac{1}{H}\int_{-D}^{\eta}(-\nabla\cdot\mathbf{u}^*\theta)dz + \frac{1}{H}\int_{-D}^{\eta}\frac{\partial}{\partial z}\left(\kappa_v\frac{\partial\theta}{\partial z}\right)dz + \frac{Q_{net}}{H\rho_{ref}C_p} + R_T, \quad (4)$$

where $\eta$ represents sea surface height, $D$ denotes a constant depth level ($D$ equals 295 m in this study), $H = D + \eta$, and $z$ is the vertical coordinate, positive upwards; $t$ is time; $\mathbf{u}$ is the three-dimensional resolved ocean velocity; $\mathbf{u}^*$ is the three-dimensional subgrid-scale velocity from the mesoscale[86] and submesoscale[87] parameterizations; $\kappa_v$ denotes the vertical diffusivity; $Q_{net}$ includes the net air-sea heat flux and internal ocean heat flux due to ice formation, $\rho_{ref}$ is the ocean reference density ($\rho_{ref} = 1026$ kg/m³), $C_p$ is the ocean heat capacity ($C_p = 3996$ J/kg/°C); $R_T$ denotes the residual. In the following discussion, the terms in the above equation are referred to as, in sequence of appearance: the total temperature tendency, contribution to total tendency from resolved ocean advection, parameterized advection,

diabatic vertical mixing, net surface heat fluxes, and residual. These terms are in units of °C/yr. Tendencies associated with lateral diffusion and KPP[63] non-local vertical mixing are not available in the simulation we use. Therefore, they are treated as a residual together with a small term, the Robert Filter tendency. These residual processes have all been saved as output in other CESM2 simulations where it has been shown that the full heat budget closes to roundoff-level error.

The decomposition of salinity tendency is similar to the heat budget analysis:

$$\frac{1}{H}\int_{-D}^{\eta}\frac{\partial S}{\partial t}dz = \frac{1}{H}\int_{-D}^{\eta}(-\nabla\cdot\mathbf{u}S)dz + \frac{1}{H}\int_{-D}^{\eta}(-\nabla\cdot\mathbf{u}^*S)dz + \frac{1}{H}\int_{-D}^{\eta}\frac{\partial}{\partial z}\left(\kappa_v\frac{\partial S}{\partial z}\right)dz + F + R_S,$$

(5)

where $F$ denotes the net surface salinity flux that is associated with precipitation, evaporation, river runoff, ice runoff, and ice melting/growth; and $R_S$ is the residual, which contains the same components as those in the heat budget analysis. Similar to the terms in the temperature budget equation, the terms in the above equation are referred to as, in sequence of appearance: the total salinity tendency, contribution to total tendency from resolved ocean advection, parameterized advection, vertical mixing, net surface salinity fluxes, and residual. These terms are in units of g/kg/yr.

This budget analysis is mainly used to investigate the differences in upper-ocean properties between the warm and cold groups. Therefore, it is conducted over a fixed depth of 295 m. An alternative approach to this analysis involves using a varying MLD. In such a case, the role of vertical mixing might be represented by the variability of MLD, without changing the conclusions of our budget analysis.

### Warming hole index
We define a warming hole index following ref. 66 to measure the response of SST to global warming at each grid cell as compared to the global mean response for each large ensemble:

$$WH_{i,j} = \left(SST_{i,j}^{2070:end} - SST_{i,j}^{begin:2000}\right) - \left(SST_{global}^{2070:end} - SST_{global}^{begin:2000}\right),$$

(6)

where $SST_{i,j}^{2070:end}$ denotes the ensemble mean DJFM SST at each grid cell, averaged from 2070 to the end of each large ensemble; $SST_{i,j}^{begin:2000}$ represents the same but averaged from the beginning of the simulations in each large ensemble to 2000; and the subscript "global" refers to a weighted average over the globe. It is notable that different large ensembles have different start and end years (Table 1). We use SST rather than the average upper-ocean temperature because the latter is not available in all large ensembles. The area-weighted averaging is conducted using the model output grid cell area for models with irregular ocean grids and the cosine of the latitude for models that provide outputs with regular ocean grids.

### Data availability
CESM2-LE is available at https://www.cesm.ucar.edu/community-projects/lens2. Simulations from CanESM2 and CSIRO-Mk3.6 are available at https://www.cesm.ucar.edu/community-projects/mmlea[52]. Simulations from MPI-ESM are available at https://esgf-data.dkrz.de/projects/mpi-ge/. Simulations from GFDL-SPEAR are available at https://www.gfdl.noaa.gov/spear_large_ensembles/. Simulations from ACCESS-ESM1-5, CanESM5, and MIROC6 are available at https://pcmdi.llnl.gov/CMIP6/.

### Code availability
The associated code is available at: https://doi.org/10.5281/zenodo.10975951.

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

## Acknowledgements

We express our gratitude to Laifang Li, Raymond G. Najjar, Jr., Isla Simpson, and Andrew M. Carleton for engaging in insightful discussions. The majority of computations for this research were performed on the Pennsylvania State University's Institute for Computational and Data Sciences' Roar supercomputer. This content is solely the responsibility of the authors and does not necessarily represent the views of the Institute for Computational and Data Sciences. We acknowledge the CESM2 Large Ensemble Community Project and supercomputing resources provided by the Institute for Basic Science Center for Climate Physics in South Korea. We also thank the US CLIVAR Working Group on Large Ensembles for providing the Multi-Model Large Ensemble Archive. This material is also based upon work supported by the National Science Foundation (NSF) National Center for Atmospheric Research (NCAR), which is a major facility sponsored by the NSF under Cooperative Agreement No. 1852977. We also acknowledge the computational resources provided by the CESM Project through the Climate Variability and Change Working Group allocation. F.C. was partially supported by the grant NA18OAR4310429 from the National Oceanic and Atmospheric Administration (NOAA), Climate Program Office (CPO), Climate Variability and Predictability Program; Modeling Analysis, Predictions, and Projections Program; and the NOAA Global Ocean Monitoring and Observing (GOMO) Program, and by the Department of Energy, Earth and Environmental System Modeling, Regional and Global Model Analysis Program. W.M.K. was partially supported by the grant 2020-0102/2040020 from the NSF GEO-NERC and by the NSF Office of Polar Programs grant OPP-2106228. E.M. was partially supported by the #NA22OAR4310111 from the NOAA CPO Modeling Analysis, Predictions, and Projections Program.

## Author contributions

Q.G., M.G., and G.D. conceived the study. Q.G. performed the analysis and wrote the first draft of the manuscript. All authors, including Q.G., M.G., G.D., W.M.K., F.C., E.M., and S.-P.X., contributed to interpreting the results and improved the manuscript.

## Competing interests

The authors declare no competing interests.
