## [Peer Review File · Nature Communications]

Wide range of possible trajectories of North Atlantic climate in a warming worldREVIEWER COMMENTS

Reviewer #1 (Remarks to the Author):

Review of “Wide range of possible trajectories of North Atlantic climate in a warming world” by Gu et al.

Climate models project a wide range of strengths of decadal temperature variability in the North Atlantic region under global warming. Narrowing the spread of climate model simulations for specific target variables and time horizons could indicate the potential for skilful climate prediction in the future. Here, the authors develop an interesting approach surrounding this idea, considering groups of ensemble members of the CESM2 model’s historical and SSP3-7.0 projection simulations that are particularly warm or cold in the subpolar North Atlantic region. Tracing back the mechanisms that differentiate these groups of ensemble members, the authors explore the physics that precondition one or the other group of North Atlantic climate trajectories, highlighting potential for decadal climate prediction along the way.

I find the study relevant and the underlying question interesting. I particularly appreciate the approach that is used as it is innovative and represents a creative use of large model ensemble simulations. The results are generally well-presented, although I have a few specific concerns about the presentation (for details see specific comments below).

While I only have minor comments on the (again, very interesting!) manuscript as is, I am wondering if the results apply widely enough to warrant publication in *Nature Communications*. The results at this stage focus on one climate model ensemble and only accomplish limited reference to “the real world” in terms of observations. This manuscript might therefore in its current form be slightly too focused on a niche audience for publication in *Nature Communications*. With the inclusion of more climate models or a stronger reference to observed climate, I can absolutely see this study become a *Nature Communications* paper. I am curious to read a reply by the authors to these concerns.

Please also consider the specific comments I outline below (line numbers would really have helped making targeted comments; I am afraid you will have to search a bit for the statements that I made reference to since I could only mark them by paragraph).

Specific comments:

Abstract Please consider specifying the SSP scenario under which SST ensemble spread more than doubles. I also think the last sentence is a bit blurry: are you insinuating that these predictions would be dynamical or statistical? I think a specification could really help the reader catch your meaning here.

Section 1, Paragraph 1 “necessitates” -> “requires”?

S1, P1 I wonder what the impacts of recent findings on the importance of volcanic eruptions on North Atlantic SST prediction skill (Swingedouw et al., 2015; Hermanson et al., 2020; Borchert et al., 2021) on your study are. Could you please comment and potentially elaborate in the manuscript?

S1, P2 Is the last sentence here redundant with the last sentence of paragraph 3?

S2.1, P1 Please specify the cutoff frequency for your lowpass filter.

S2.1, P1 Please provide a reference for the statement “wider range of possible future NNA climate states” (wider than what?).

S2.1, P1 When stating that the subpolar North Atlantic is one of the hot spots of decadal climate prediction in current climate, consider discussing the potentially strong role of natural forcing and its implications for your study (Hermanson et al., 2021; Borchert et al., 2021).

S2.1, P2 I would have appreciated a bit more introduction to the rationale behind the research approach taken in this study. Please consider elaborating on why you do what you do, and what the anticipated outcome is.

S2.1, P2 “greater (less)” -> “greater (smaller)”

S2.1, P2 The decision to consider a simulation warm or cold based on one year in a five year time span appears arbitrary to me. What is the sensitivity of your results to this choice?

S2.1, P2 How do the different sample sizes between the warm and cold groups affect your results? Could the larger amount of warm members compared to cold ones be related to a (non-linear, since you detrend linearly I think) global warming effect?

S2.1, P3 “group that starts” -> “group that start”

Figure 1 caption The title says “ensemble mean” but you show full ensembles mainly. Consider removing the “mean” in the figure caption title.

S2.2, P1 I struggle to see how you find that the rate of deep convection shutdown is crucial in shaping NNASST trajectories. Why is this factor more important than the others? Please elaborate.

S2.2, P2 Please explain variable symbols where you use them, not only in the Methods section.

S2.2, P3 Is the salinity budget shown in Figure 3d? Generally: the figure panels are referenced in the text in seemingly random order, not in the order in which they appear. That does not make it easier to follow the story.

S2.2, P4 Do you mean AMOC by “large-scale ocean circulation”? Please specify if that is the case. The statement is otherwise ambiguous and not particularly strong.

S2.2, P5 I am not sure I find it proper that the authors gloss over the strength of the residual term in the temperature tendency. Please interpret this finding: its influence is as strong as that of vertical mixing!

S2.3, P2 I am not sure I understand the first sentence of this paragraph. What do you mean by “abnormal”?

S2.3, P4 It would be interesting to read a bit more on why you think the AMOC drives the cooling of the North Atlantic relative to the warm group here.

S2.3, P4 I also wonder about the cause and effect relationship between the warm and cold groups and the composite plots (Fig. 6) discussed in this paragraph. Can you make any statements on whether the sea ice cover is low because the NNASST is warmer or vice versa? If so, how?

S3 I find this a relatively weak discussion section. There is summary, but barely any discussing of the results. How do your results fit into the published literature on similar topics and the fate of the North Atlantic? Have you answered your research questions? What are the specific answers?

S4, P2 Interesting that the CESM2-LE is started from four different AMOC states. Do the warm and cold groups that you define here cluster around these AMOC states? Is there a preference for members started from a particularly strong/weak AMOC to fall into either group?

References

Borchert et al. (2021) <https://doi.org/10.1029/2020GL091307>
Hermanson et al. (2020) <https://doi.org/10.1029/2019JD031739>
Swingedouw et al. (2015) <https://doi.org/10.1038/ncomms7545>

Reviewer #2 (Remarks to the Author):

This manuscript analyzes the CESM2 LE evolution of SST in the North Atlantic, describing two sets based on the 21st century SST evolution: a COLD set and a WARM set. The difference is traced back to differences in Labrador Sea convection and associated feedback mechanisms. The two sets of ensemble members diverge in the middle of the 21st century as convection shuts down in the COLD set, and subsequently converge again once convection in the WARM set also winds down.

I very much enjoyed reading this manuscript. It is well written and the figures illustrate the text nicely. I have a few minor comments that might strengthen the manuscript, but they are not crucial:

- In the budget analysis (Fig. 4), the terms that contribute to the difference between the WARM and COLD sets by far are 'diabetic vertical mixing' and 'residual terms'. I believe both of these are really just deep convection, where 'diabetic vertical mixing' is the direct component, and the residual containing the non-local KPP contributions as mentioned in section 4. If this is correct, it would be good that state that more explicitly.
- The feedback mechanisms are presented as novel finding, but previous studies have identified similar mechanisms, if not always identical, based on shorter time scales but events that already happened (e.g. a sea ice feedback in the deep convection event of 2008 (Våge et al., 2009, Surprising return of deep convection to the subpolar North Atlantic Ocean in winter 2007-2008, *Nature Geoscience*, 2, 67-72, doi: 10.1038/NGEO382)). It may be worth putting the findings in that perspective, as it lends credence to the feedbacks happening in the real world.
- I am really curious what the difference in evolution of latent heat flux is between the COLD and WARM sets. I would expect the latent heat flux to be a lot smaller in the COLD set, but the 'surface flux' term in the budgets causes either warming and salinification in the COLD set, or less cooling and freshening. The latter would be consistent with a smaller latent heat flux in terms of temperature, but not the salinity. It's a combination of all surface fluxes of course, but I would find it instructive to see them split into their various components, and also to see the signs of the individual terms per set and not just the difference (can go into the supplementary material of course).
- Last, since the authors suggest that monitoring in the coming years can identify the trajectory the real ocean system is on, it would be helpful if they gave some specific recommendations on which quantities should be measured, where, at what frequency, etc. That would make the implications of this manuscript more actionable.

Wide range of possible trajectories of North Atlantic climate in a warming world

We deeply appreciate the constructive feedback provided by the two reviewers. We have modified the manuscript, figures, and supplementary materials as they suggested and we believe these changes improved the manuscript overall.

1 Response to Reviewer 1

1.1 Summary

Climate models project a wide range of strengths of decadal temperature variability in the North Atlantic region under global warming. Narrowing the spread of climate model simulations for specific target variables and time horizons could indicate the potential for skilful climate prediction in the future. Here, the authors develop an interesting approach surrounding this idea, considering groups of ensemble members of the CESM2 model’s historical and SSP3-7.0 projection simulations that are particularly warm or cold in the subpolar North Atlantic region. Tracing back the mechanisms that differentiate these groups of ensemble members, the authors explore the physics that precondition one or the other group of North Atlantic climate trajectories, highlighting potential for decadal climate prediction along the way.

I find the study relevant and the underlying question interesting. I particularly appreciate the approach that is used as it is innovative and represents a creative use of large model ensemble simulations. The results are generally well-presented, although I have a few specific concerns about the presentation (for details see specific comments below).

While I only have minor comments on the (again, very interesting!) manuscript as is, I am wondering if the results apply widely enough to warrant publication in *Nature Communications*. The results at this stage focus on one climate model ensemble and only accomplish limited reference to “the real world” in terms of observations. This manuscript might therefore in its current form be slightly too focused on a niche audience for publication in *Nature Communications*. With the inclusion of more climate models or a stronger reference to observed climate,

I can absolutely see this study become a *Nature Communications* paper. I am curious to read a reply by the authors to these concerns.

Please also consider the specific comments I outline below (line numbers would really have helped making targeted comments; I am afraid you will have to search a bit for the statements that I made reference to since I could only mark them by paragraph).

We have followed your suggestion and added the line number to the manuscript.

Thank you for the suggestions to include more climate models. We have now incorporated the results of seven additional large ensembles into the new Section 2.4. We demonstrated that in models exhibiting the formation of the North Atlantic warming hole (indicative of deep convection shutdown and AMOC slowdown), there is an increase in subpolar North Atlantic SST ensemble spread in the 21st century. This finding is consistent with what we have identified in CESM2-LE and further corroborates our findings. However, due to the limited variables available from these large ensembles, we were unable to investigate the mechanisms within each large ensemble.

Please refer to the newly added Figures 8 and 9, and Section 2.4 for details. We have also revised the abstract (Lines 29-30) and the Methods section (Sections 4.2 and 4.5, including a new Table 1) to reflect the additional investigation of other large ensembles.

1.2 Specific comments

1. Abstract

Please consider specifying the SSP scenario under which SST ensemble spread more than doubles. I also think the last sentence is a bit blurry: are you insinuating that these predictions would be dynamical or statistical? I think a specification could really help the reader catch your meaning here.

We appreciate your comment. We have added the SSP scenario information on Lines 19-23: “Using the Community Earth System Model 2 Large Ensemble under historical and the Shared Socioeconomic Pathway 3-7.0 future radiative forcing scenarios, we uncover that the ensemble spread in northern North Atlantic sea surface temperature (SST) more than doubles during the mid-21st century, highlighting an exceptionally wide range of possible climate states.”

We have refined the last sentence of the abstract for greater specificity as follows: “By monitoring these mechanisms in real time and extending dynamical model predictions after positive feedbacks activate, we may achieve skillful long-lead North Atlantic decadal predictions that are effective for multiple decades.”

Additionally, we have detailed the potential approaches for achieving accurate long-lead predictions in the Discussion Section (Lines 365-393) as follows:

“In the CESM2-LE, we identify distinct trajectories starting around 2030 that lead to either warm or cold NNASST, which then lasts for decades until the end of the

simulations. This suggests a potential increase in the multidecadal predictability of North Atlantic SST, despite increased internal variability. Such added predictability on decadal timescales over the upcoming 50-70 years could hold significant socioeconomic value, informing climate adaptation and mitigation strategies. Here, we propose two potential avenues for future work that can leverage this enhanced predictability in reality.

The first avenue involves the comparison between observations and the model simulations. The distinct trajectories between the warm and cold groups in CESM2-LE can be identified in multiple variables (e.g., NNASST, MLD, NHT, AMOC) starting from ~ 2030 — only a few years from now. However, with the exception of SST, the majority of these do not have sufficiently long records in the observations. Monitoring the NNASST in the observations over the upcoming decade and comparing this to the envelop of NNASST in the CESM2-LE warm and cold groups may allow us to infer whether the observations are consistently within one trajectory or the other. From this, we may infer the North Atlantic climate state in the following 50-70 years.

The second avenue for future work is related to dynamical model predictions. Current decadal prediction models generally provide predictions out to about 10 years [55, 56]. The extended predictability revealed here suggests that these models could potentially offer skillful predictions over longer periods once the mean climate allows the activation of positive feedbacks. The differing timing of the increase in subpolar SST standard deviation suggests that the timing of crossing the “tipping point” can vary across different models. Our analysis with CESM2-LE indicates that the divergence in trajectories is triggered around 2027 in this model, pinpointing this year as the approximate time when the mean state crosses the “tipping point” that activates the positive feedbacks. Thus, we may extend the prediction period beyond the typical 10 years for dynamical predictions initialized around 2027 in the CESM2 to achieve skillful long-lead predictions.”

2. Section 1, Paragraph 1

“necessitates” -> “requires”?

We have followed this suggestion and made the modification on Line 37.

3. S1, P1

I wonder what the impacts of recent findings on the importance of volcanic eruptions on North Atlantic SST prediction skill (Swingedouw et al., 2015; Hermanson et al., 2020; Borchert et al., 2021) on your study are. Could you please comment and potentially elaborate in the manuscript?

Thank you for sharing these references about how volcanic eruptions can improve the North Atlantic SST prediction skill. These references show that volcanic eruptions can affect climate variability and predictability in the North Atlantic, particularly through their effects on the NAO and AMOC. These mechanisms are closely relevant to our study. Therefore, just like internal variability, we suggest that the impact of volcanic eruptions on the climate might also vary over time under global warming.

The focus of this study is on internal variability and anthropogenic external forcing. To complement the perspective of natural external forcing in the manuscript, we added the following sentences on Lines 394-399 in the Discussion section with references: “In addition to the anthropogenic external forcing and internal variability focused on in this study, previous studies have shown that natural external forcing such as volcanic eruptions can also facilitate skillful decadal SST prediction in the North Atlantic through dynamical mechanisms involving NAO and AMOC [70-72]. Thus, another avenue for future work is to explore how the climate impacts and predictability associated with natural external forcing may change under global warming.”

4. S1, P2

Is the last sentence here redundant with the last sentence of paragraph 3?

To better connect this paragraph with the following one and avoid redundancy, we have modified the sentence as follows: “However, what remains an open question is how anthropogenic external forcing may modulate internal variability in the North Atlantic on decadal timescales.”

5. S2.1, P1

Please specify the cutoff frequency for your lowpass filter.

We have specified the lowpass filter on Lines 82-84: “Decadal variability in North Atlantic SST is isolated by applying a lowpass filter (11-year running mean) to winter (December-January-February-March, DJFM) North Atlantic SST for each member of the CESM2-LE.”

6. S2.1, P1

Please provide a reference for the statement “wider range of possible future NNA climate states” (wider than what?).

Thank you for the suggestion. We have added a reference to Lines 84-88: “We find an increased ensemble spread across the 100 ensemble members during the mid-21st century that is concentrated north of 50°N (see Supplementary Movie 1), indicating a wider range of possible future northern North Atlantic climate states compared to those in the historical period.”

7. S2.1, P1

When stating that the subpolar North Atlantic is one of the hot spots of decadal climate prediction in current climate, consider discussing the potentially strong role of natural forcing and its implications for your study (Hermanson et al., 2021; Borchert et al., 2021).

In our response to your comment #3, we have added discussions on the significant influence of natural external forcing on North Atlantic decadal prediction, elaborating on its relevance to our study on Lines 394-399. We hope that this addition addresses your comment. The sentence you referred to is designed to clarify our interest in this region: given its high predictability under current climate conditions, we are interested

in exploring potential changes in predictability under future climate conditions. Consequently, we chose not to elaborate on factors affecting North Atlantic predictability, such as natural external forcing, in this paragraph.

8. S2.1, P2

I would have appreciated a bit more introduction to the rationale behind the research approach taken in this study. Please consider elaborating on why you do what you do, and what the anticipated outcome is.

Thank you for the suggestion. Accordingly, we have revised this paragraph to more clearly articulate the rationale behind the research approach employed in this study (Lines 95-99): “This increase in North Atlantic internal variability might be stochastic in nature and thus unpredictable, or it could stem from processes that, when traced back, may provide some predictability. To identify the onset of the broadened SST distribution and any potential precursors, we classify ensemble members into warm and cold groups, then conduct composite analyses of these groups.”

9. S2.1, P2

“greater (less)” -> “greater (smaller)”

We have made the modification on Line 102 according to your suggestion. Thank you.

10. S2.1, P2

The decision to consider a simulation warm or cold based on one year in a five year time span appears arbitrary to me. What is the sensitivity of your results to this choice?

To answer this question, we conducted multiple sensitivity analyses on how we classify the warm and cold groups. In the current manuscript, we classify these groups by comparing the deviation of the NNASST index from the ensemble mean in any year of the 5 years with the highest NNASST ensemble standard deviation to the piControl standard deviation. To test the sensitivity of this choice, we modified this 5-year period to either a 1-year period (Fig. **R1a**) or a 9-year period (Fig. **R1b**). The averaged NNASST index in both the warm and cold groups remains similar to that with the current classification method.

Furthermore, we explored an alternative classification method that involves selecting an equal number of ensemble members into the warm and cold groups based on their NNASST index, averaged over a number of years. For instance, in Fig. **R2a**, we calculated the average NNASST over the 5 years with the highest standard deviation. Based on this average, we selected the top 10 and bottom 10 ensemble members for the warm and cold groups, respectively. In Figs. **R2b-f**, we examined the sensitivity of the classification to both the number of ensemble members in each group and the number of years used for averaging. Generally, the magnitude of the NNASST difference between the warm and cold groups depends on the ensemble size, but the shape of the NNASST averaged in both groups is comparable to that in Fig. 1a. This further suggests that our results are robust to different classification methods.

To reflect these sensitivity tests, we have added the statement “Our results are also robust to different classification methods (not shown).” to Line 107.

Fig. R1. Sensitivity test on the classification criteria of the warm and cold groups. **a**, same as Fig. 1a, but the warm and cold groups are classified based on their winter NNASST during the year with the highest NNASST standard deviation. Ensemble members are assigned to the warm (cold) group if their NNASST index is greater (smaller) than or equal to the ensemble mean plus (minus) one standard deviation of the NNASST in the CESM2 piControl simulation during that year. **b**, same as **a**, but the warm and cold groups are classified based on any year of the 9 years with the highest NNASST standard deviation.

11. S2.1, P2

How do the different sample sizes between the warm and cold groups affect your results? Could the larger amount of warm members compared to cold ones be related to a (non-linear, since you detrend linearly I think) global warming effect?

Based on the answer to the previous question, selecting a same number of ensemble members into the warm and cold groups leads to a similar result to our current classification method with different sample sizes (Fig. **R2**). This indicates that the different sample sizes do not affect our results. In addition, this sample difference is considered in the significance test used in the study.

In the original classification method, we classified the warm and cold groups based on the deviation of the NNASST from the ensemble mean. Theoretically, with the ensemble mean removed, the effect of external forcing is considered to be excluded. A different number of members being classified into the warm and cold groups indicates that the distribution of NNASST deviations from the ensemble mean is skewed. We agree with your assumption that this could be associated with some non-linear effects that are superimposed on the internal variability due to global warming.

12. S2.1, P3

“groups that starts” -> “groups that start”

Thank you for catching this. We have modified it on Line 113.

Fig. R2. Sensitivity test on the classification criteria of the warm and cold groups. Same as Fig. 1a, but an equal number of ensemble members are classified into the warm and cold groups based on the average NNASST index over a number of years with the highest NNASST standard deviation. **a**, top and bottom 10 members based on the 5 years with the highest NNASST standard deviation. **b**, top and bottom 20 members based on the 5 years with the highest NNASST standard deviation. **c**, top and bottom 30 members based on the 5 years with the highest NNASST standard deviation. **d**, top and bottom 40 members based on the 5 years with the highest NNASST standard deviation. **e**, top and bottom 40 members based on the 10 years with the highest NNASST standard deviation. **f**, top and bottom 40 members based on the 20 years with the highest NNASST standard deviation.

13. Figure 1 caption

The title says “ensemble mean” but you show full ensembles mainly. Consider removing the “mean” in the figure caption title.

To avoid confusion, we have modified the title of the caption for Figure 1 to “Lowpass-filtered DJFM time series averaged in the warm (orange) and cold (blue) groups.”

14. S2.2, P1

I struggle to see how you find that the rate of deep convection shutdown is crucial in shaping NNASST trajectories. Why is this factor more important than the others? Please elaborate.

We understand your concern. What we intend to convey is that, as discussed in the previous section, the difference in MLD shutdown rates precedes the differences in AMOC, NHT, and NNASST. This prompts us to further investigate the cause of the difference in MLD shutdown rates between the groups, with the aim of identifying the trigger for this cascade of differences.

To clarify this point, we have modified Lines 151-153 to read: “Given that the difference in MLD shutdown rates between the groups precede the differences in AMOC, NHT, and NNASST, it is crucial to investigate what causes the MLD difference between the groups.”

15. S2.2, P2

Please explain variable symbols where you use them, not only in the Methods section.

Thank you for the suggestion. We added the following sentence on Lines 162-165: “Here, α_θ and β_S represent the thermal expansion coefficient and haline contraction coefficient, respectively, while θ and S denote potential temperature and salinity (see Methods for details).”

16. S2.2, P3

Is the salinity budget shown in Figure 3d? Generally: the figure panels are referenced in the text in seemingly random order, not in the order in which they appear. That does not make it easier to follow the story.

We understand your concern. The original Fig. 3d depicted the ensemble mean salinity and its vertical gradient. To save space, it was combined with Figures 3a-c. We have now rearranged the order of the figures by moving the original Figure 3d to become the new Figure 7. After this rearrangement, the figure panels are referenced in the correct sequence. It is worth noting that subplots e-g in Figure 1 are referenced later in the manuscript than a-d. They are included in Figure 1 because 1) they represent the same type of figures as a-d, and 2) including them in Figure 1 makes it easier for readers to compare the sequence of different events depicted in these subplots.

17. S2.2, P4

Do you mean AMOC by “large-scale ocean circulation”? Please specify if that is the case. The statement is otherwise ambiguous and not particularly strong.

You correctly understood that we meant “AMOC” in this context. We have now specified this on Lines 196-197: “These processes reduce the NNASST in the cold group through changes in AMOC.”

18. S2.2, P5

I am not sure I find it proper that the authors gloss over the strength of the

residual term in the temperature tendency. Please interpret this finding: its influence is as strong as that of vertical mixing!

The residual term is indeed large because it encompasses important processes for this region. Unfortunately, these are not output by the model and must therefore be computed as a residual. To help clarify this, we have expanded the discussion regarding the residual terms when presenting both the salinity and heat budget analysis. Specifically, on Lines 188-191 we added: “This term encompasses lateral diffusion and K-Profile vertical mixing Parameterization (KPP; [63]) non-local vertical mixing that are not available in the simulation employed in this study, as well as a small term, the Robert Filter tendency.” On Lines 212-214 we added: “Similar to the salinity budget analysis, this term mainly consists of lateral diffusion and KPP non-local vertical mixing that are not available in CESM2-LE.”

We are not able to separate the lateral diffusion and KPP non-local vertical mixing because they are not available in the simulation we use. Therefore, the discussion cannot be extended further. The description of the residual term was originally only in the Methods Section. Now, we have repeated this information when presenting the budget analysis for clarity. Thank you for bringing this concern to our attention.

19. S2.3, P2

I am not sure I understand the first sentence of this paragraph. What do you mean by “abnormal”?

By “not abnormal”, we mean that a similar difference in stochastic NAO variability occurs in other time periods. To clarify this, we have modified the sentence to “This difference in stochastic NAO variability is a common occurrence within the period of study and it only persists for a short time period.”, thereby avoiding the use of “abnormal”. To better support this argument, we have expanded the discussion by providing an actual example on Lines 231-236 as follows: “Specifically, a comparable difference in the NAO index is evident around 1935 between the two groups, even with the current classification (Fig. 1e). This NAO difference, spanning approximately a decade, leads to a significant difference in NNASST for multiple years around 1950 between the groups (Fig. 1a). However, this difference in NNASST is relatively short-lived and does not intensify, in contrast to the divergence starting from 2035.”

20. S2.3, P4

It would be interesting to read a bit more on why you think the AMOC drives the cooling of the North Atlantic relative to the warm group here.

Previous literature has shown that the AMOC is a key component in driving poleward heat transport in the North Atlantic. This argument was embedded in Section 2.1 on Lines 123-126: “In addition to the transport from the Labrador Sea, the AMOC plays a dominant role in poleward heat transport [e.g., 40], which can be tied back to deep convection in the Labrador Sea in CESM [20, 35, 58, 59].” To make this information more explicitly stated, we modified the introduction (Lines 58-59) by adding “The AMOC is a key driver of poleward heat transport in the North Atlantic Ocean [40-42].”

In Section 2.1, we show that the differences in AMOC and northward heat transport precede the differences in NNASST between the warm and cold groups. Based on this information and previous literature cited in this manuscript, we argue in Section 2.3 that the AMOC is one of the drivers of the stronger cooling of the North Atlantic in the cold group compared to the warm group. To clarify this, we added the figure reference on Lines 253-255: “Simultaneously, the entire North Atlantic basin cools relative to the warm group because of large-scale ocean circulation (e.g., AMOC, Fig. 1c) and transport originating from the Labrador Sea (Fig. 2). ”

21. S2.3, P4

I also wonder about the cause and effect relationship between the warm and cold groups and the composite plots (Fig. 6) discussed in this paragraph. Can you make any statements on whether the sea ice cover is low because the NNASST is warmer or vice versa? If so, how?

Thank you for the question. The cause and effect relationship between SST and sea ice is not very clear. Originally, we argued that the difference in AMOC and associated northward heat transport, as well as the difference in Labrador Sea SST, would lead to the sea ice difference. However, from another perspective, with more sea ice coverage in the cold group, the albedo would be larger, reflecting more solar radiation back, which may further decrease the SST. This is the so-called ice-albedo feedback. To clarify this, we modified Lines 255-258 to “These processes contribute to reduced sea ice melting in the cold group compared to the warm group (Fig. 6a), and the positive ice-albedo feedback [64] may further amplify the differences in SST and sea ice between the groups. ”

22. S3

I find this a relatively weak discussion section. There is summary, but barely any discussing of the results. How do your results fit into the published literature on similar topics and the fate of the North Atlantic? Have you answered your research questions? What are the specific answers?

To make the discussion section more comprehensive, we added discussion on how our results fit into the published literature on similar topics on Lines 341-348: “Previous studies have offered valuable insights into the mechanisms driving North Atlantic SST variability using observations, piControl simulations, and historical simulations with limited realizations [29-35]. In this study, we utilize a single model initial-condition large ensemble containing 100 simulations that can allow us to study internal variability within the context of changes in external forcing. This addresses a gap left by previous studies, allowing us to diagnose how anthropogenic external forcing can modulate internal variability and how their interactions may improve predictability on decadal timescales. ”

Our research question is “how future climate change might impact decadal variability in the North Atlantic Ocean”, and the second paragraph in the Discussion section summarized our answer to this question.

In terms of the fate of the North Atlantic, we show a drastic increase of the range of

potential climate states in the northern North Atlantic in the mid-21st century. However, as we mentioned in the manuscript, with the understanding of the mechanisms we propose in this study, we may improve the predictability in this region. To clarify how this improvement could be achieved and to make the Discussion section more comprehensive, we added Lines 365-393. Please refer the answer to your Comment #1, or the Discussion section for more details.

Furthermore, a brief summary and discussion of the results from the seven additional models are added on Lines 358-364: “Additional investigation of seven other large ensembles shows that this increase in internal SST variability broadly exists in models that simulate the formation of a North Atlantic warming hole under global warming, despite variations in timing across different models. While there exist debates, recent studies have provided evidence for the slowdown of the AMOC and the formation of the North Atlantic warming hole in the observations [44, 67-69]. This evidence underscores the potential real-world manifestation of our findings, warranting further attention from our community.”

23. S4, P2

Interesting that the CESM2-LE is started from four different AMOC states. Do the warm and cold groups that you define here cluster around these AMOC states? Is there a preference for members started from a particularly strong/weak AMOC to fall into either group?

To answer this question, we present Fig. **R3**, which shows the average NNASST within four groups of ensemble members. These groups are categorized based on their initialization with maximum, decreasing, minimum, and increasing AMOC states, respectively, with each group containing 20 ensemble members. There are no significant differences between the various groups of ensemble members during the mid-21st century. This indicates that the warm and cold groups do not cluster around these AMOC states.

References

- Borchert et al. (2021) <https://doi.org/10.1029/2020GL091307>
Hermanson et al. (2020) <https://doi.org/10.1029/2019JD031739>
Swingedouw et al. (2015) <https://doi.org/10.1038/ncomms7545>

2 Response to Reviewer #2

2.1 Summary

This manuscript analyzes the CESM2 LE evolution of SST in the North Atlantic, describing two sets based on the 21st century SST evolution: a COLD set and a WARM set. The difference is traced back to differences in Labrador Sea convection and associated feedback mechanisms. The two sets of ensemble members diverge in the middle of the 21st century as convection shuts down in

Fig. R3. DJFM NNASST in ensemble members initialized with different AMOC states. The blue, purple, yellow, and red lines represent the 20-member average NNASST initialized with maximum, decreasing, minimum, and increasing AMOC states, respectively. The shadings, using the same colors as their corresponding lines, represent the ensemble spread, calculated as the ensemble mean \pm one ensemble standard deviation, for the 20-member groups. Black dots on the lower axis signify years when the ensemble means between members initialized with maximum and minimum AMOC states are significantly different at the 95% level based on a two-tailed Student's t test.

the COLD set, and subsequently converge again once convection in the WARM set also winds down.

I very much enjoyed reading this manuscript. It is well written and the figures illustrate the text nicely. I have a few minor comments that might strengthen the manuscript, but they are not crucial:

2.2 Specific comments

1. In the budget analysis (Fig. 4), the terms that contribute to the difference between the WARM and COLD sets by far are ‘diabatic vertical mixing’ and ‘residual terms’. I believe both of these are really just deep convection, where ‘diabatic vertical mixing’ is the direct component, and the residual containing the non-local KPP contributions as mentioned in section 4. If this is correct, it would be good that state that more explicitly.

Thank you for the suggestion. The residual term contains 1) KPP non-local vertical mixing, 2) lateral diffusion, and 3) a small term, the Robert Filter tendency. Because 1) and 2) are not available in the simulations we use, it is not possible to separate them. Therefore, we cannot understand the residual term as being solely related to deep convection.

To clarify this, we have expanded the discussion regarding the residual terms when presenting both the salinity and heat budget analysis. Specifically, on Lines 188-191 we added: “This term encompasses lateral diffusion and K-Profile vertical mixing Parameterization (KPP; [63]) non-local vertical mixing that are not available in the simulation employed in this study, as well as a small term, the Robert Filter tendency.” On Lines 212-214 we added: “Similar to the salinity budget analysis, this term mainly consists of lateral diffusion and KPP non-local vertical mixing that are not available in CESM2-LE.”

- 2. The feedback mechanisms are presented as novel finding, but previous studies have identified similar mechanisms, if not always identical, based on shorter time scales but events that already happened (e.g. a sea ice feedback in the deep convection event of 2008 (Våge et al., 2009, Surprising return of deep convection to the subpolar North Atlantic Ocean in winter 2007-2008, Nature Geoscience, 2, 67-72, doi: 10.1038/NGEO382)). It may be worth putting the findings in that perspective, as it lends credence to the feedbacks happening in the real world.**

We appreciate the reference you provided. In Våge et al., 2009, it was observed that the anomalously widespread pack ice in the Labrador basin allowed the cold air from the continent to remain largely unaffected by air-sea heat flux as it moved towards the interior basin, where convection took place. This contributed to anomalously cold SST and thus strengthened deep convection in winter 2007-2008. In their study, the increased heat flux from the ocean to the atmosphere is key to the resumption of deep convection. Therefore, the sea ice feedback discussed in their study is associated with cold air temperature, which eventually leads to increased heat loss from the ocean to the atmosphere. In our study, the difference in salinity, rather than temperature, between the warm and cold groups is the dominant factor contributing to density differences (Fig. 3). Thus, the feedback #2 associated with sea ice is different from what they have demonstrated in their study.

Inspired by this question, we realized that we can add another realistic feedback mechanism associated with sea ice that may occur under the conditions described in our study, namely, the ice-albedo feedback. The cold group has more ice cover, which would reflect more solar radiation and lead to less melting. This feedback may further amplify the differences in SST and sea ice coverage between the two groups. Since this is a well-known feedback mechanism, we briefly mention it alongside the feedback we propose on Lines 255-258: “These processes contribute to reduced sea ice melting in the cold group compared to the warm group (Fig. 6a), and the positive ice-albedo feedback [64] may further amplify the differences in SST and sea ice between the groups.”

- 3. I am really curious what the difference in evolution of latent heat flux is**

between the COLD and WARM sets. I would expect the latent heat flux to be a lot smaller in the COLD set, but the ‘surface flux’ term in the budgets causes either warming and salinification in the COLD set, or less cooling and freshening. The latter would be consistent with a smaller latent heat flux in terms of temperature, but not the salinity. It’s a combination of all surface fluxes of course, but I would find it instructive to see them split into their various components, and also to see the signs of the individual terms per set and not just the difference (can go into the supplementary material of course).

We understand your concern. To address this question, we have added a new Supplementary Fig. 3 that shows a decomposition of both the surface salinity flux and the surface temperature flux as differences between the cold and warm groups (cold - warm). We have also included Fig. R4 for you that provides this for each group (i.e. not the cold and warm group difference). As depicted in these two figures, the cold group experiences less evaporation compared to the warm group during the period of interest, leading to a reduced salinity tendency (fresher) and an increased temperature tendency (warmer). Thus, the contribution of evaporation opposes the surface salinity flux but aligns with the surface temperature flux, which is consistent with your assumption. The reason why the contribution of evaporation opposes the total surface salinity flux is due to the role of both precipitation and melting, as shown in Supplementary Fig. 3.

To incorporate the information associated with the new Supplementary Fig. 3 into the manuscript, we have modified Lines 180-182 and 215-218. Lines 180-182 now read: “Surface flux counteracts the total salinity tendency difference, primarily due to reduced melting and precipitation in the cold group relative to the warm group (Supplementary Fig. 3a).” Lines 215-218 now read: “In contrast, surface heat flux consistently dampens the difference between the groups starting from around 2025, which can be primarily attributed to the reduced turbulent heat flux from the ocean to the atmosphere in the cold group (Supplementary Fig. 3b). ”

- 4. Last, since the authors suggest that monitoring in the coming years can identify the trajectory the real ocean system is on, it would be helpful if they gave some specific recommendations on which quantities should be measured, where, at what frequency, etc. That would make the implications of this manuscript more actionable.**

Thank you for the suggestion. We have modified the Discussion section to incorporate a more actionable approach. In the original approach we proposed, many key variables that can be monitored to determine the trajectory of the real ocean system might not be available in the observations. Therefore, in the revised Discussion section, we clarify this issue and introduce another approach that could potentially facilitate long-lead decadal predictions.

This part of the Discussion section now reads:

“In the CESM2-LE, we identify distinct trajectories starting around 2030 that lead to either warm or cold NNASST, which then lasts for decades until the end of the

Fig. R4. Salinity and heat budget analysis in the upper 295 m of the Labrador Sea: Decomposition of the surface flux. **a**, the warm (solid lines) and cold (dashed lines) group averaged total surface salinity flux (g/kg/yr; blue), as well as contributions to the surface flux from runoff (yellow), ice runoff (purple), freshwater fluxes between the ocean and sea-ice models due to processes such as sea ice and snow melting (grey), precipitation (cyan), evaporation (orange), salt exchange between the ocean and sea-ice models due to the salinity of sea ice (dark green), frazil ice formation (maroon). **b**, the warm (solid lines) and cold (dashed lines) group averaged total surface temperature flux (°C/yr; blue), as well as contributions to the surface flux from shortwave radiation (yellow), ice runoff (purple), heat flux exchange between the ocean and sea-ice models (grey), sensible heat flux (cyan), evaporation (orange), snow (pink), upwelling longwave radiation (dark green), downwelling longwave radiation (light green), frazil ice formation (maroon).

simulations. This suggests a potential increase in the multidecadal predictability of North Atlantic SST, despite increased internal variability. Such added predictability on decadal timescales over the upcoming 50-70 years could hold significant socioeconomic value, informing climate adaptation and mitigation strategies. Here, we propose two potential avenues for future work that can leverage this enhanced predictability in reality.

The first avenue involves the comparison between observations and the model simulations. The distinct trajectories between the warm and cold groups in CESM2-LE can be identified in multiple variables (e.g., NNASST, MLD, NHT, AMOC) starting from ~ 2030 — only a few years from now. However, with the exception of SST, the majority of these do not have sufficiently long records in the observations. Monitoring the NNASST in the observations over the upcoming decade and comparing this to the envelop of NNASST in the CESM2-LE warm and cold groups may allow us to infer whether the observations are consistently within one trajectory or the other. From this, we may infer the North Atlantic climate state in the following 50-70 years.

The second avenue for future work is related to dynamical model predictions. Current decadal prediction models generally provide predictions out to about 10 years [55, 56]. The extended predictability revealed here suggests that these models could potentially offer skillful predictions over longer periods once the mean climate allows the activation of positive feedbacks. The differing timing of the increase in subpolar SST standard deviation suggests that the timing of crossing the “tipping point” can vary across different models. Our analysis with CESM2-LE indicates that the divergence in trajectories is triggered around 2027 in this model, pinpointing this year as the approximate time when the mean state crosses the “tipping point” that activates the positive feedbacks. Thus, we may extend the prediction period beyond the typical 10 years for dynamical predictions initialized around 2027 in the CESM2 to achieve skillful long-lead predictions.”

REVIEWERS' COMMENTS

Reviewer #1 (Remarks to the Author):

Second Review for "Wide range of possible trajectories of North Atlantic climate in a warming world"

I appreciate the additional work that the authors have done to accommodate both reviewers' concerns. Their thorough revision has impressively demonstrated the robustness of the effect that the authors put forward, using multiple large ensembles and additional analysis. This complementary analysis has convinced me that this manuscript is indeed fit for publication in Nature Communications.

I have no further concerns and congratulate the authors on an important research paper.

All the best with the further process,

Leonard Borchert

Reviewer #2 (Remarks to the Author):

The authors have addressed my suggestions and I am happy for this manuscript to be published.